# Drug–Drug Interactions between Tamsulosin and Mirabegron in Healthy Individuals Do Not Affect Pharmacokinetics and Hemodynamic Parameters Significantly

**DOI:** 10.3390/ph16101457

**Published:** 2023-10-13

**Authors:** Wonsuk Shin, A-Young Yang, Hyounggyoon Yoo, Anhye Kim

**Affiliations:** 1Department of Clinical Pharmacology and Therapeutics, CHA Bundang Medical Center, CHA University School of Medicine, Seongnam 13520, Gyeonggi-do, Republic of Korea; wonsug89@chamc.co.kr (W.S.); adud610@gmail.com (A.-Y.Y.); hgyoo0317@cha.ac.kr (H.Y.); 2Institute for Biomedical Informatics, CHA University School of Medicine, Seongnam 13488, Gyeonggi-do, Republic of Korea

**Keywords:** drug–drug interaction, tamsulosin, mirabegron, pharmacokinetics

## Abstract

Overactive bladder (OAB) is characterized by urinary urgency and increased urinary frequency, substantially affecting quality of life. Tamsulosin and mirabegron combination therapy has been studied as a safe and effective treatment option for patients with OAB. This study evaluated the effects of combining these two drugs on their pharmacokinetics and safety profiles in healthy Korean males. In this open-label, fixed-sequence, three-period, drug–drug interaction phase 1 study, a total of 36 male participants were administered multiple doses of tamsulosin alone (0.2 mg once daily), mirabegron alone (50 mg once daily), or a combination of both drugs. The results showed that the combination of tamsulosin and mirabegron increased tamsulosin exposure in the plasma by approximately 40%. In contrast, the maximum plasma concentration of mirabegron was reduced by approximately 17% when administered with tamsulosin. No clinically significant changes in the safety profiles, vital signs, or clinical laboratory test results were observed in this study. In conclusion, there were no clinically relevant drug–drug interactions between tamsulosin and mirabegron in terms of pharmacokinetics, safety, and tolerability, suggesting that their combination could be a promising treatment option for patients with OAB.

## 1. Introduction

Overactive bladder (OAB) is a condition characterized by urinary urgency, which involves a sudden and intense need to urinate, accompanied by a sensation of bladder irritation, without urinary tract infection or any other obvious cause. It commonly involves both a high urinary frequency (>8 times) during the daytime and nocturia (excessive urination at night) [1,2]. The prevalence of OAB in the population aged ≥65 years is 30.9% [3,4]. The condition has a detrimental effect on physical activity, psychosocial well-being, and quality of life [5]. The treatment plan for OAB varies depending on the severity of the symptoms and the patient’s health status; common treatment options include behavioral therapy, pharmacotherapy, and surgical treatment.

Pharmacotherapy, along with behavioral therapy, is internationally recognized as the primary treatment for OAB to relieve symptoms and reduce urge incontinence. It involves treatment with anticholinergics and/or β3-adrenergic receptor agonists. Anticholinergic drug therapy is the first-line treatment to help relax the bladder muscle and reduce the urgency and frequency of urination by blocking the action of acetylcholine, a neurotransmitter that stimulates bladder contractions. Some examples in this class include oxybutynin, solifenacin, tolterodine, and trospium [6,7]. β3-adrenergic receptor agonists, such as mirabegron, function by stimulating beta-3 adrenergic receptors in the bladder, resulting in the relaxation of the bladder muscle and an increase in its capacity to hold urine. β3-adrenergic receptor agonists are used as an alternative to anticholinergic drugs for OAB treatment, especially in patients who cannot tolerate or do not respond well to anticholinergics [8]. Several studies have reported significant therapeutic effects of monotherapy using both classes of medications [9,10,11]. However, oral antimuscarinics, commonly used as first-line treatment and in monotherapy, have high discontinuation rates because of bothersome side effects or inadequate clinical response [12,13,14,15]. To improve therapeutic efficacy, clinicians either increase the dose of the drug [16], switch to a different antimuscarinic, or try a combination of antimuscarinics, resulting in higher rates of side effects [17]. In cases where medication does not provide satisfactory treatment, procedures such as magnetic stimulation, bladder distension, alcohol injection, botulinum toxin injection, urinary diversion, augmentation cystoplasty, and neuromodulation are performed [9,18]. Given that these procedures are more invasive or inconvenient treatment options [18], combination pharmacotherapy may offer an additional promising non-invasive therapeutic management step between single-agent pharmacotherapy and more invasive approaches for the treatment of patients with OAB [19]. In May 2018, the Food and Drug Administration (FDA) approved the combination therapy of a β3-adrenergic agonist and an antimuscarinic, solifenacin, as a pharmacological treatment option for OAB [20].

Mirabegron, the first β3 adrenergic receptor agonist approved by the FDA for OAB treatment, has gradually gained acceptance in recent years as most clinical trials have reported better pharmacological profiles and improved patient compliance compared to those for antimuscarinics [21,22,23]. Tamsulosin, a selective α1 adrenergic receptor antagonist approved for the treatment of benign prostatic hyperplasia (BPH) [24], enhances bladder storage function by inhibiting the C-fibers in the urethra and alleviates the response of sensory nerves to bladder stimulation signals and pain. Thus, its efficacy and effectiveness in treating OAB have been evaluated [25,26]. In recent clinical trials, tamsulosin and mirabegron combination therapy for OAB showed improvements in alleviating symptoms without causing additional side effects [27,28]. Consequently, tamsulosin and mirabegron combination therapy is likely to be actively pursued as an alternative to conventional drugs for patients who cannot continue with existing drug therapies due to insufficient efficacy or drug-related side effects.

To validate and provide a basis for mirabegron and tamsulosin combination therapy, the potential drug–drug interactions (DDIs), based on the main metabolic pathway and excretion routes of each drug, should be considered. Mirabegron is metabolized extensively via various mechanisms including phase 1 and 2 metabolism. Moreover, it functions as a moderate cytochrome P450 (CYP) 2D6 (CYP2D6) inhibitor, as evidenced by an increase in the exposure of CYP2D6 substrates in several DDI studies [29,30]. Tamsulosin is primarily metabolized in the liver by cytochrome P450 3A4 (CYP3A4) and CYP2D6 [31]. Considering the metabolism of both drugs, there is the possibility of a CYP2D6-mediated DDI. Thus, this study aimed to evaluate the effects of DDIs between tamsulosin and mirabegron on their pharmacokinetics and safety profiles in healthy Korean male participants.

## 2. Results

### 2.1. Demographic Characteristics

A total of 36 male participants were enrolled in the study after providing informed consent, of whom 2 withdrew from the study. An analysis of the demographic characteristics and safety was conducted for the 36 participants who received at least one dose of the investigational product (IP). The mean and standard deviation (in parentheses) of the age, height, weight, and body mass index (BMI) of the participants were 29.00 (7.93) y, 176.17 (6.35) cm, 77.01 (10.55) kg, and 24.76 (2.73) kg/m^2^, respectively.

### 2.2. Pharmacokinetic Analysis

The pharmacokinetic set consisted of the 34 participants who underwent all blood samplings required to calculate the pharmacokinetic parameters. The pharmacokinetic parameters for tamsulosin and mirabegron when administered alone or in combination are summarized in Table 1. The mean plasma concentration–time plots for tamsulosin and mirabegron are presented in Figure 1. The profiles of the major pharmacokinetic parameters, the maximum plasma concentration at steady state (C_max,ss_), and the area under the concentration–time curve within a dosing interval at steady state (AUC_τ,ss_), for each participant, are presented in Figure 2.

The comparison of treatment with tamsulosin alone (T1) and tamsulosin combined with mirabegron (T3) revealed that both treatments showed similar time–concentration graphs, indicating a multiphasic elimination phase after reaching peak plasma concentrations at 5 h (Figure 1A). However, the mean C_max,ss_ and AUC_τ,ss_ for tamsulosin were 9.53 ng/mL and 101.96 h·ng/mL in the T1 group, whereas they were 6.70 ng/mL and 73.63 h·ng/mL, respectively, in the T3 group. Moreover, the C_max,ss_, AUC_τ,ss_, and minimum plasma concentration at steady state (C_min,ss_) were significantly increased, and the apparent clearance at steady state (CL_ss_/F) and apparent volume of distribution at steady state (Vd_ss_/F) were significantly decreased in the T3 group compared to those in the T1 group. The geometric mean ratios (GMRs) and 90% confidence intervals (CIs; in parentheses) for the C_max,ss_ and AUC_τ,ss_ values of co-treatment with tamsulosin and mirabegron to those of tamsulosin alone were 1.4018 (1.3127–1.4969) and 1.3609 (1.2786–1.4486), respectively (Table 1). In T1, the C_max,ss_ and AUC_τ,ss_ for tamsulosin increased by about 1.4 times compared to those in T3 (Figure 2A1,A2).

The comparison of treatment with mirabegron alone (T2) and tamsulosin combined with mirabegron (T3) revealed that both treatments showed similar time–concentration graphs, indicating a multiphasic elimination phase after reaching peak plasma concentrations at 5 h (Figure 1B). For groups T2 and T3, the mean C_max,ss_ was 39.50 and 33.48 ng/mL, the mean AUC_τ,ss_ was 282.75 and 277.63 h·ng/mL, and the mean t_1/2_ was 19.72 and 19.81 h, respectively. There was no significant difference in the pharmacokinetic parameters between the treatments. The GMRs (90% CIs) for the C_max,ss_ and AUC_τ,ss_ values of treatment with tamsulosin combined with mirabegron to those of treatment with mirabegron alone were 0.8367 (0.7439–0.9410) and 0.9761 (0.9189–1.0369), respectively (Table 1). Both C_max,ss_ and AUC_τ,ss_ decreased slightly in the combination treatment (T3) compared to those in T2; however, these changes were not clinically significant (Figure 2B1,B2).

### 2.3. Safety Analysis

Safety analysis was performed on the 36 participants who received the IPs at least once. A total of 19 treatment-emergent adverse events (TEAEs) occurred in 10 participants (27.78%): 5 TEAEs in 5 participants (13.89%) undergoing treatment with T1, 8 TEAEs in 5 participants (14.29%) undergoing treatment with T2, and 6 TEAEs in 5 participants (14.71%) undergoing treatment with T3. All TEAEs were mild and resolved spontaneously without any medications, except for one TEAE (otitis media), which was resolved after treatment with antibiotics and painkillers. The details of the TEAEs are summarized in Table 2. Among the TEAEs, 11 TEAEs occurring in 8 participants (22.22%) were assessed as adverse drug reactions (ADRs): 3 ADRs in 3 participants (13.89%) in T1, 3 ADRs in 3 participants (14.29%) in T2, and 5 ADRs in 5 participants (14.71%) in T3. There were no significant differences in the incidence of TEAEs and ADRs among the treatments (*p* = 0.9952 and *p* = 0.6696, respectively). In addition, the most common ADRs included headache, dizziness, and retrograde ejaculation (two participants each); retrograde ejaculation was reported only for T3.

Vital signs and changes from the baseline measured on the screening day were evaluated, wherein the vital sign data were measured pre-dose on the last IP administration day for each treatment (days 5, 19, and 26 in T1, T2, and T3, respectively) (Table 3). In all three treatments, the vital signs were within the normal range, and no significant difference was observed between the treatments regarding changes from the baseline.

No clinically significant changes were observed in the clinical laboratory test results (except for TEAEs), chest X-ray examinations, ECG measurements, or physical examinations.

## 3. Discussion

The present study was designed as a randomized, open-label, multiple-dose, fixed-sequence, three-period, three-treatment study to evaluate the DDI between mirabegron and tamsulosin. As tamsulosin is primarily metabolized by CYP3A4 and CYP2D6, and mirabegron is a moderate inhibitor of CYP2D6, DDIs were evaluated in steady state after the administration of multiple doses to maximize the chance of identification. In contrast, considering that this study was conducted in healthy volunteers, it was designed in a fixed sequence to minimize the duration of drug exposure from a risk–benefit perspective. This study was well-designed to evaluate the effects of DDI between the two IPs on their pharmacokinetics and safety.

The results of this study revealed that the combination of tamsulosin and mirabegron increased tamsulosin exposure (AUC_τ,ss_, C_max,ss_) by approximately 40% and decreased tamsulosin clearance (CL_ss_/F) by approximately 35%, while slightly decreasing the C_max,ss_ for mirabegron by approximately 17% compared to that under mirabegron monotherapy. According to the classification of CYP inhibitors by the FDA [32], a moderate inhibitor increases the AUC_τ,ss_ of a CYP sensitive index substrate by 2–5-fold. In this study, the AUC_τ,ss_ for tamsulosin increased by approximately 1.4-fold only when combined with mirabegron, a moderate inhibitor of CYP2D6. This result could be attributed to the extensive metabolism of tamsulosin, mainly by CYP3A4 and CYP2D6, resulting in a slightly smaller-than-expected decrease in tamsulosin clearance [31,33]. In contrast, mirabegron did not show any pharmacokinetic changes under co-administration with tamsulosin other than a slight decrease in C_max,ss_ by approximately 17%, which is consistent with the findings of a previous study [34] and is considered clinically insignificant [32].

In the present study, tamsulosin and mirabegron were administered during the fasting state. According to the prescribing information for each drug, tamsulosin is recommended to be taken approximately half an hour following a meal [33], whereas mirabegron can be taken with or without food [35], based on the results of their food effect studies. Although mirabegron exposure decreases with meals, it has demonstrated both safety and efficacy irrespective of food contents and intake [35]. The C_max,ss_ for tamsulosin, related to orthostasis, decreases when it is administered in the fed state; thus, it is recommended to be taken after meals. In clinical settings, postprandial administration is reasonable when both drugs are taken in combination, and it is speculated that the magnitude of C_max,ss_ increase observed in this study would be reduced.

Notably, no serious adverse events or clinically significant safety issues were observed in this study. In addition, given that mirabegron, a beta-3 agonist, may increase heart rate and blood pressure, while tamsulosin may cause hypotension by alpha-a1 receptor-mediated vasodilation, the blood pressure and heart rate at a steady state during each treatment were analyzed. However, cardiovascular interactions were not observed, and the characteristics of headache and dizziness, the most common ADRs, were also unrelated to postural change. Thus, cardiovascular interactions between these two drugs were not clinically relevant in this study with a low therapeutic dose in healthy volunteers, which is similar to the results of a previous study with a higher therapeutic dose (tamsulosin 0.4 mg, mirabegron 100 mg) in middle-aged to older adult men [34]. Moreover, two cases of retrograde ejaculation, which is a well-known side effect of tamsulosin that is exacerbated in a dose-dependent manner [33], were observed in the combination treatment. However, as the frequency of incidence and the sample size in this study were too small, it is difficult to infer the association between its incidence and DDIs.

There are some limitations to this study. First, 0.2 mg tamsulosin was administered in this study, whereas tamsulosin 0.4 mg once daily is the recommended dose for the treatment of the symptoms of BPH. However, according to previous studies [27,36] in which tamsulosin and mirabegron were co-administered for patients with OAB, the recommended dose was 0.2 mg tamsulosin and 50 mg mirabegron. Although FDA guidelines recommend conducting drug interaction studies at the maximum approved dose [32], owing to the dose-proportional pharmacokinetics exhibited by tamsulosin [37], we believe that conducting the study at a dose of 0.2 mg is also adequate to meet the study objectives. Therefore, the results of this study provide substantive evidence for pharmacokinetic DDIs. Second, this study was conducted on young male healthy volunteers, but the target condition, OAB, is prevalent in the older adult population, especially women. Mirabegron has been reported to have no significant impact by age, but body-weight-corrected systemic exposure increases by approximately 20–30% in female patients; however, no dose adjustment is necessary for older adults, regardless of their sex [35]. In contrast, it is suggested that the pharmacokinetics and dispositions of tamsulosin could exhibit a modest prolongation in older adult men compared to young ones. The intrinsic clearance of tamsulosin has been noted to decline concomitantly with advancing age, culminating in a notable 40% elevation in the total systemic exposure (signified by AUC_τ,ss_) among individuals aged 55–75 years compared to those aged 20–32 years. Nevertheless, it is noteworthy that no discernible differences in terms of overall safety or efficacy were perceptible between the older and younger study participants. In addition, another study evaluating the effects of DDIs between mirabegron and tamsulosin in healthy middle-aged to older adult men demonstrated the absence of clinically relevant changes in cardiovascular safety or safety profiles [34]. Therefore, our results support the findings of previous studies.

## 4. Materials and Methods

### 4.1. Participants and Study Design

This study was conducted at the Global Clinical Trials Center of the CHA Bundang Medical Center, CHA University, Seongnam, Republic of Korea, with strict adherence to the key ethical principles outlined in the Declaration of Helsinki, the Good Clinical Practice Guidelines of the International Council for Harmonization of Technical Requirements for Pharmaceuticals for Human Use, and local laws and regulations. The study protocol was reviewed and approved by the MFDS and Institutional Review Board (IRB) of CHA University (IRB no. CHAMC 2020-04-051). Furthermore, the study was registered and can be found on ClinicalTrials.gov (https://clinicaltrials.gov, accessed on 24 July 2021), with the identifier NCT04485585.

Before the commencement of the clinical trial, the study information and other relevant details were provided to all participants. Screening procedures were conducted exclusively on individuals who voluntarily consented to participate in the clinical trial. The inclusion criteria focused on healthy males aged 19–55 years, who were assessed for their eligibility based on their medical history, vital signs, and the results of physical examination, clinical laboratory tests, and a 12-lead electrocardiogram (ECG). The exclusion criteria included individuals with a clinically significant medical history, those who had participated in other clinical trials within the last six months before the screening, and those who were taking prescribed medications that could not be temporarily discontinued for at least two weeks before the screening.

We conducted a randomized, open-label, multiple-dose, fixed-sequence, three-period, three-treatment study (Figure 3). The IPs were tamsulosin HCl (0.2 mg, Hanmi Pharmaceutical Co., Ltd., Seoul, Republic of Korea) and mirabegron (50 mg, Astellas Pharma Korea, Inc., Seoul, Republic of Korea). Eligible participants were divided into fixed-sequence groups and administered 0.2 mg tamsulosin HCl once daily for 5 d (T1). After a 5-day washout period, the participants were administered 50 mg mirabegron once daily for 11 d (T2). Next, the participants were further administered 50 mg mirabegron and 0.2 mg tamsulosin HCl for 5 d (T3). Considering a previous study that included 48 participants (24/arm) to evaluate the pharmacokinetic drug interaction between mirabegron and tamsulosin [34], the required number of participants was set to 36, considering the dropout rate and to improve the validity of the clinical data. The 5-day washout period between periods 1 and 2 was more than 5 times the terminal half-life of tamsulosin (8.85 ± 2.98 h) [38]. In contrast, there was no washout period between periods 2 and 3 to maintain a steady mirabegron concentration. During each period, the participants were administered either tamsulosin, mirabegron, or their combination with 150 mL of water during the fasting state. Blood samples were collected at the following time points: pre-dose on days 1, 3, and 4, pre-dose (0 h) and 1, 2, 3, 4, 5, 6, 8, 10, 12, and 24 h post-dose on day 5 for tamsulosin; pre-dose on days 11, 17, and 18, pre-dose (0 h) and 1, 2, 3, 4, 5, 6, 8, 10, 12, and 24 h post-dose on day 19 for mirabegron; pre-dose on days 24 and 25 (tamsulosin only), pre-dose (0 h) and 1, 2, 3, 4, 5, 6, 8, 10, 12, and 24 h post-dose on day 26 for tamsulosin and mirabegron.

### 4.2. Blood Sampling and Determination of Tamsulosin and Mirabegron Plasma Concentrations

Blood samples were collected in heparinized tubes at each blood sampling time point, and the plasma was separated through centrifugation at 1900× *g* for 10 min at 4 °C. The collected plasma samples were transferred to 2 Eppendorf tubes at a volume of approximately ≥1.0 mL and stored in a freezer at ≤−70 °C, until further analysis.

The plasma concentrations of tamsulosin and mirabegron were determined through liquid chromatography (tamsulosin: Exion LC AB SCIEX, Washington, DC, USA; mirabegron: Shimadzu Prominence ultra-fast LC, Shimadzu, Kyoto, Japan) combined with tandem mass spectrometry (tamsulosin: API 4000, AB SCIEX; mirabegron: QTRAP 6500+, AB SCIEX), based on validated analytical procedures adopted by the FDA [39] and the Korean Ministry of Food and Drug Safety [40]. For tamsulosin, the calibration curves were linear in the range of 0.1–50 ng/mL (correlation coefficient, r > 0.9950), with a lower limit of quantification (LLOQ) of 0.1 ng/mL. Tamsulosin-d_4_ HCl and rac-Mirabegron-d_5_ were used as internal standards. The assay range for mirabegron was 0.2–200 ng/mL (r > 0.9950), with an LLOQ of 0.2 ng/mL. The accuracy of the assay was within the range of 98.5–105.8% for tamsulosin, and 87.0–111.3% for mirabegron. The precision coefficients of variation for tamsulosin and mirabegron were <6.0% and <13.5%, respectively.

### 4.3. Pharmacokinetic Assessment

Non-compartmental analysis was performed using Phoenix WinNonlin software version 8.2 (Certara Co., Princeton, NJ, USA) to determine the following pharmacokinetic parameters for tamsulosin and mirabegron: C_max,ss_, C_min,ss_, time to reach C_max,ss_ (T_max,ss_), AUC_τ,ss_, elimination half-life at steady state (t_1/2_), CL_ss_/F, and Vd_ss_/F. Plasma drug concentration–time profiles are presented in linear and log-transformed scales. C_max,ss_, C_min,ss_, and T_max,ss_ were measured, and AUC_τ,ss_ was calculated using the linear trapezoidal linear interpolation method. This method involves linear trapezoidal summation when the blood concentration increases and log-linear trapezoidal summation when it decreases. The elimination rate constant (k_e_) was estimated by performing a linear regression analysis on the data points included in the terminal phase of the log-linear plot of the concentration–time data, and the t_1/2_ was calculated from the ratio of the natural logarithm of 2 and k_e_. CL_ss_/F was calculated as dose/AUC_τ_, and Vd_ss_/F was calculated as (CL_ss_/F)/k_e_.

### 4.4. Safety Assessment

Safety was evaluated based on TEAEs, vital signs, and the results of physical examination, 12-lead ECGs, and clinical laboratory tests. TEAEs were either spontaneously reported by the participants or identified through the data collected during scheduled interviews throughout the study period. Vital signs, including systolic blood pressure, diastolic blood pressure, and heart rate, were measured at the baseline of each visit (0 h), except for visits involving IP administration, as these were measured before IP administration. These measurements were performed with the participant maintaining a stable supine position for at least 3 min, ensuring no sudden positional changes. All TEAEs were coded according to the Medical Dictionary for Regulatory Activities version 23.0 and summarized based on treatment, severity, and association with tamsulosin and mirabegron.

### 4.5. Statistical Analysis

Descriptive statistics were used to summarize baseline demographics, such as age, weight, height, and BMI. Pharmacokinetic parameters and safety profiles were also evaluated using descriptive statistics. All statistical analyses were conducted using SAS version 9.4 (SAS Institute, Inc., Cary, NC, USA). Primary pharmacokinetic endpoints (C_max,ss_ and AUC_τ,ss_) were log-transformed to develop a mixed-effects model with the treatment effect as the fixed effect and participant effect as the random effect. GMRs with 90% CIs of the primary pharmacokinetic parameters for tamsulosin alone vs. tamsulosin and mirabegron and for mirabegron alone vs. tamsulosin and mirabegron were estimated to evaluate the pharmacokinetic drug interactions. Numerical data for these two treatments were compared using the independent *t*-test or Mann–Whitney U test, and those for all treatments were compared via one-way analysis of variance (ANOVA). The Scheffe method was used for the post hoc analysis, along with ANOVA. Categorical data were compared using the Chi-squared or Fisher’s exact test.

## 5. Conclusions

In conclusion, no clinically relevant DDIs regarding the pharmacokinetics, safety, or tolerability between tamsulosin and mirabegron were observed. Combination therapy with these two drugs could contribute to synergistic effects due to their differing mechanisms and increased compliance.

## Figures and Tables

**Figure 1 pharmaceuticals-16-01457-f001:**
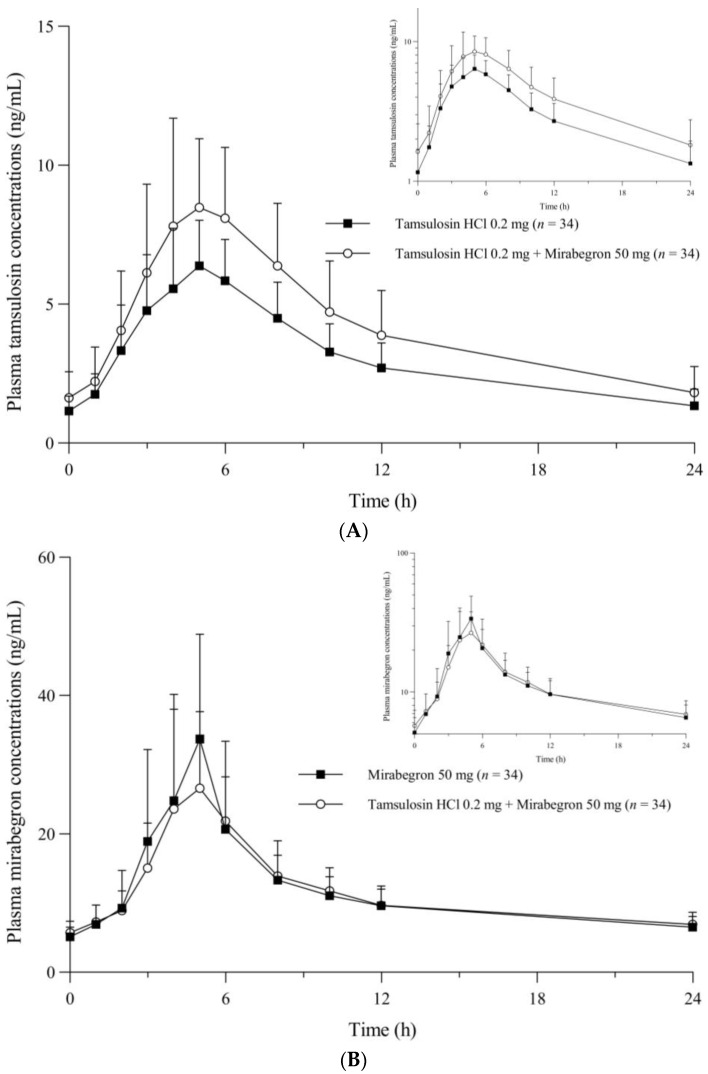
The mean plasma concentration–time plots for (**A**) tamsulosin and (**B**) mirabegron.

**Figure 2 pharmaceuticals-16-01457-f002:**
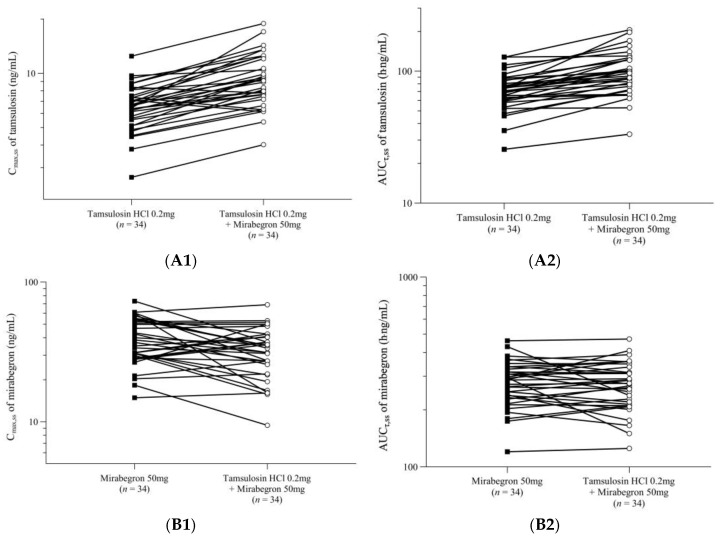
Participant profiles: (**A1**) C_max,ss_ and (**A2**) AUC_τ,ss_ after the administration of tamsulosin, and (**B1**) C_max,ss_ and (**B2**) AUC_τ,ss_ after the administration of mirabegron.

**Figure 3 pharmaceuticals-16-01457-f003:**
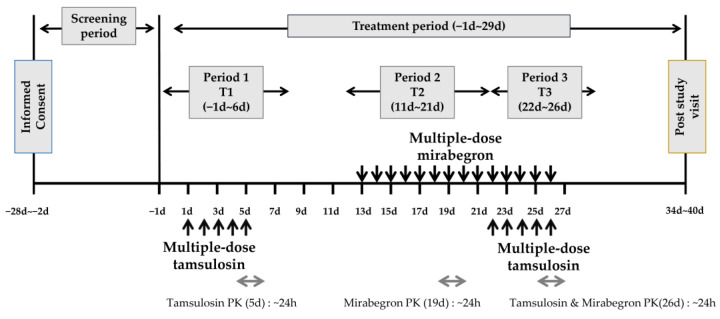
Study schema.

**Table 1 pharmaceuticals-16-01457-t001:** Pharmacokinetic parameters for tamsulosin 0.2 mg HCl with and without 50 mg mirabegron at steady state and for 50 mg mirabegron with and without tamsulosin 0.2 mg HCl at steady state.

Treatment	C_max,ss_ (ng/mL)	AUC_τ,ss_ (h·ng/mL)	T_max,ss_ (h)	t_1/2_ (h)	C_min,ss_ (ng/mL)	CL_ss_/F (L/h)	Vd_ss_/F (L)
Tamsulosin							
Tamsulosin alone	6.70 (1.89)	73.63 (22.54)	5.00 (3.00–6.00)	10.98 (2.6)	1.15 (0.53)	2.77 (1.09)	42.02 (12.75)
Tamsulosin + Mirabegron	9.53 (3.31)	101.96 (38.36)	5.00 (3.00–6.00)	10.20 (2.11)	1.59 (0.93)	2.06 (0.86)	28.85 (8.2)
GMR (90% CI)	1.4018 (1.3127–1.4969)	1.3609 (1.2786–1.4486)					
Intra-participant CV (%)	16.1	15.29					
*p*-values ^(1)^	0.0001	0.0005	0.4517	0.1769	0.0214	0.0039	<0.0001
Mirabegron							
Mirabegron alone	39.50 (14.37)	282.75 (73.91)	5.00 (2.00–6.00)	19.72 (3.24)	5.11 (1.44)	190.65 (59.36)	5440.18 (1900.02)
Tamsulosin + Mirabegron	33.48 (12.92)	277.63 (78.61)	4.53 (3.00–6.00)	19.81 (5.4)	5.70 (1.67)	196.40 (63.77)	5696.03 (2781.96)
GMR (90% CI)	0.8367 (0.7439–0.9410)	0.9761 (0.9189–1.0369)					
Intra-participant CV (%)	29.23	14.8					
*p*-values	0.0739	0.7829	0.308	0.9333	0.1192	0.7016	0.6593

^(1)^ Independent *t*-test was used to assess significant differences between the treatments. Data are presented as mean and standard deviation (in parentheses), except for T_max_, which is represented by the median (range) value. AUC_τ,ss_, area under the concentration–time curve within a dosing interval at steady state; CI, confidence interval; CL_ss_/F, apparent clearance at steady state; C_max,ss_, the maximum plasma concentration at steady state; C_min,ss_, the minimum plasma concentration at steady state; CV, coefficient of variation; HCl, hydrochloride; GMR; geometric mean ratio; t_1/2_, elimination half-life at steady state (t_1/2_ was derived using data observed solely during the dosing interval); T_max,ss_, time to reach C_max,ss_; Vd_ss_/F, apparent volume of distribution at steady state.

**Table 2 pharmaceuticals-16-01457-t002:** Summary of treatment-emergent adverse events (TEAEs).

TEAE	Tamsulosin Alone (T1) (*n* = 36)	Mirabegron Alone (T2) (*n* = 35)	Tamsulosin + Mirabegron (T3) (*n* = 34)	Total (*n* = 36)
Subjects with at least one TEAE	5 (13.89) [5]	5 (14.29) [8]	5 (14.71) [6]	10 (27.79) [19]
				*p* = 0.9952 ^(1)^
Otitis media			1 (2.94) [1]	1 (2.78) [1]
Alanine aminotransferase increased	3 (8.33) [3]	2 (5.71) [2]		3 (8.33) [5]
Aspartate aminotransferase increased		1 (2.86) [1]		1 (2.78) [1]
Blood creatine phosphokinase increased	1 (2.78) [1]	3 (8.57) [3]		3 (8.33) [4]
Hemoglobin increased		1 (2.86) [1]		1 (2.78) [1]
Dizziness		1 (2.86) [1]	1 (2.94) [1]	2 (5.56) [2]
Headache	1 (2.78) [1]		2 (5.88) [2]	3 (8.33) [3]
Retrograde ejaculation			2 (5.88) [2]	2 (5.56) [2]

^(1)^ Fisher’s exact test was used to assess significant differences in the incidence of TEAEs. Values are presented as the number of participants (percentage of all participants) [number of TEAEs]. TEAEs were identified using the preferred term as per MedDRA version 23.0.

**Table 3 pharmaceuticals-16-01457-t003:** Summary and comparison of vital signs among the treatments.

Vital Sign	Timepoint of Measurement ^(1)^				
	Screening (*n* = 36)	Tamsulosin Alone (T1) (*n* = 35)	Mirabegron Alone (T2) (*n* = 34)	Tamsulosin + Mirabegron (T3) (*n* = 34)	*p*-Value ^(2)^
SBP (mmHg)	128.97 (12.04)	119.23 (8.92)	120.56 (9.56)	117.18 (8.72)	0.3058
DBP (mmHg)	78.78 (8.25)	76.26 (9.71)	73.76 (6.77)	75.29 (8.61)	0.4708
HR (bpm)	77.08 (9.47)	64.97 (9.28)	67.97 (8.82)	67.00 (8.32)	0.5490
	Change from the Screening in each treatment				
SBP (mmHg)	-	−10.57 (12.99)	−9.06 (11.22)	−12.44 (12.24)	0.5120
DBP (mmHg)	-	−2.83 (10.34)	−5.06 (7.65)	−3.53 (10.84)	0.6243
HR (bpm)	-	−12.26 (10.84)	−12.44 (11.27)	−10.41 (10.07)	0.6898

^(1)^ Measured at 0 h (pre-dose) in the steady state of each period: day 5 in tamsulosin alone (T1), day 19 in mirabegron alone (T2), and day 26 in the combination (T3). ^(2)^ The differences among the 3 treatments were compared using one-way analysis of variance. Data from the screening were excluded from the statistical analysis. Data are presented as mean (standard deviation). DBP, diastolic blood pressure; HR, heart rate; SBP, systolic blood pressure.

## Data Availability

The data are not publicly available due to confidentiality reasons.

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
