# Peer review of "Drug–Drug Interactions between Tamsulosin and Mirabegron in Healthy Individuals Do Not Affect Pharmacokinetics and Hemodynamic Parameters Significantly"

_pharmaceuticals, 2023, doi:10.3390/ph16101457_

Round 1

Reviewer 1 Report

This manuscript was well written to evaluate the effects of combining these two drugs on their pharmacokinetics and safety profiles in 15 healthy Korean males. The title and abstract clearly mention the study goal and are found appropriate. Although this work is interesting and relevant for the preclinical study, some issues need to be resolved:

1.      What IS was used when determination of tamsulosin and mirabegron plasma concentrations.

2.      The FDA guideline on method validation should be detailed, considering cite some references.

3.      Why the dose of tamsulosin HCl (0.2 mg), and mirabegron (50 mg)?

Author Response

Reviewer #1:

This manuscript was well written to evaluate the effects of combining these two drugs on their pharmacokinetics and safety profiles in 15 healthy Korean males. The title and abstract clearly mention the study goal and are found appropriate. Although this work is interesting and relevant for the preclinical study, some issues need to be resolved:

Comment 1: What IS was used when determination of tamsulosin and mirabegron plasma concentrations.

Response 1: Thank you for reviewing our manuscript and for your insightful and helpful comments. We used Tamsulosin-d4 HCl and rac-Mirabegron-d5 as internal standards. We have accordingly revised the sentence on line 324 to clarify this.

Comment 2: The FDA guideline on method validation should be detailed, considering cite some references.

Response 2: Thank you for your suggestion. We have accordingly cited the guidelines for validation by the FDA and the Korean Ministry of Food and Drug Safety on line 327.

Comment 3: Why the dose of tamsulosin HCl (0.2 mg), and mirabegron (50 mg)?

Response 3: As we already described in the discussion section (lines 244–255), we determined the dose by referring to previous studies recommending a dose of 0.2 mg and 50 mg of tamsulosin and mirabegron, respectively, for co-administration in patients with OAB. Although the FDA guidelines recommend conducting drug interaction studies at the maximum approved dose, owing to the dose-proportional pharmacokinetics exhibited by tamsulosin, we believe that conducting the study at 0.2 mg is adequate to meet the study objectives. We have revised the main text to incorporate this information as follows:

Lines 244–253:

“First, 0.2 mg tamsulosin was administered in this study, whereas tamsulosin 0.4 mg once daily is the recommended dose for the treatment of the symptoms of BPH. However, according to previous studies [27,36], in which tamsulosin and mirabegron were co-administered for patients with OAB, the recommended dose was 0.2 mg tamsulosin and 50 mg mirabegron. Although FDA guidelines recommend conducting drug interaction studies at the maximum approved dose [32], owing to the dose-proportional pharmacokinetics exhibited by tamsulosin [37], we believe that conducting the study at a dose of 0.2 mg is also adequate to meet the study objectives. Therefore, the results of this study provide substantive evidence for pharmacokinetic DDIs

Reviewer 2 Report

The article is really good, but data were not well evaluated. The role article did not presented statistical evaluation about data obtained.

At table 1, all the data should be analyzed by  statistical test, e not mentioned at discussion : results are 2 or 3 times higher.

Authors should describe explore better the reason of higher AUC of Tamsulosin, in combination, not followed by higher values of half-life.

Besides, how was the pharmacokinetic calculation done? This is also not described.

Author Response

Reviewer #2:

The article is really good, but data were not well evaluated. The role article did not presented statistical evaluation about data obtained.

Comment 1: At table 1, all the data should be analyzed by statistical test, e not mentioned at discussion : results are 2 or 3 times higher. 

Response 1: Thank you for reviewing our manuscript and for your insightful and helpful comments. As per your suggestion, we added the statistical analysis in table 1 and described it in the results and discussion sections as follows:

Lines 108–114:

“However, the mean Cmax,ss and AUCτ,ss for tamsulosin were 9.53 ng/mL and 101.96 h·ng/mL in the T1 group, whereas they were 6.70 ng/mL and 73.63 h·ng/mL, respectively, in the T3 group. Moreover, the Cmax,ss, AUCτ,ss, and the minimum plasma concentration at steady state (Cmin,ss) were significantly increased, and the apparent clearance at steady state (CLss/F) and apparent volume of distribution at steady state (Vdss/F) were significantly decreased in the T3 group compared to those in the T1 group.”

Lines 122–125:

“For groups T2 and T3, the mean Cmax,ss was 39.50 and 33.48 ng/mL, the mean AUCτ,ss was 282.75 and 277.63 h·ng/mL, and the mean t1/2 was 19.72 and 19.81 h, respectively. There was no significant difference in the pharmacokinetic parameters between the treatments.”

Lines 207–216:

“The results of this study revealed that the combination of tamsulosin and mirabegron increased tamsulosin exposure (AUCτ,ss, Cmax,ss) by approximately 40% and decreased tamsulosin clearance (CLss/F) by approximately 35%, while slightly decreasing the Cmax,ss for mirabegron by approximately 17% compared to mirabegron monotherapy. According to the classification of CYP inhibitors by the FDA (32), a moderate inhibitor increases the sensitive index CYP substrate by 2–5-fold. In this study, the AUCτ,ss for tamsulosin increased by approximately 1.4-fold only when combined with mirabegron, a moderate inhibitor of CYP2D6. This result could be attributed to the extensive metabolism of tamsulosin, mainly by CYP3A4 and CYP2D6, resulting in a slightly smaller-than-expected decrease in tamsulosin clearance.”

Comment 2: Authors should describe explore better the reason of higher AUC of Tamsulosin, in combination, not followed by higher values of half-life.

Response 2: Thank you for your important comment. According to FDA guidelines (Clinical Drug Interaction Studies — Cytochrome P450 Enzyme- and Transporter-Mediated Drug Interactions Guidance for Industry), pharmacokinetic parameters are evaluated using AUCτ,ss in multiple-dose studies. Therefore, we collected blood samples were collected up to 24 h after the administration of the last dose during each period. As tamsulosin exhibits a multiphasic elimination phase, the terminal elimination phase could not be fully observed within this 24-h timeframe, which might explain why there was no change in the half-life of tamsulosin after co-administration of mirabegron in this study.

Comment 3: Besides, how was the pharmacokinetic calculation done? This is also not described.

Response 3: As suggested by the reviewer, we have added a more detailed description of the linera trapezoidal linear interpolation method for calculating the AUC in the revised manuscript as follows (lines 337–342):

“Plasma drug concentration-time profiles are presented in linear and log-transformed scales. Cmax,ss, Cmin,ss, and Tmax,ss were measured, and AUCτ,ss was calculated using the linear trapezoidal linear interpolation method. This method involves linear trapezoidal summation when the blood concentration increases and log-linear trapezoidal summation when it decreases.

Reviewer 3 Report

Thank you for the opportunity to review your manuscript, Drug-Drug interactions between Tamsulosin and Mirabegron in healthy individuals do not affect pharmacokinetics and hemodynamic parameters significantly.

This is a well-structured study report and addresses an important clinical question for the understanding of drug-drug interactions.

I have some questions about drug determination that could enhance the clarity of your study report:

·         Has the analytical method been validated? It is not clear from the text (lines 310-311).

·         Has an internal standard been used? What has been used?

·         How long were the samples in the freezer? Was the stability of these samples determined at -70°C?

Author Response

Reviewer #3:

This is a well-structured study report and addresses an important clinical question for the understanding of drug-drug interactions.

I have some questions about drug determination that could enhance the clarity of your study report:

Comment 1: Has the analytical method been validated? It is not clear from the text (lines 310-311). 

Response 1: Thank you for reviewing our manuscript and for your insightful and helpful comments. We performed the validation according to the guidelines for validation by the FDA and the Korean Ministry of Food and Drug Safety. The referenced guidelines have been accordingly cited on line 324.

Comment 2: Has an internal standard been used? What has been used?

Response 2: We used Tamsulosin-d4 HCl and rac-Mirabegron-d5 as internal standards. We have accordingly specified this on line 327.

Comment 3: How long were the samples in the freezer? Was the stability of these samples determined at -70°C?

Response 3: Thank you for your question. All samples were stored at -70°C or below and analyzed within 4 months from collection. Unfortunately, there were no published studies evaluating the stability of tamsulosin and mirabegron at -70°C. Instead, we refer to articles that showed reliable stability at -50°C and -40°C for over 1 month, respectively.

1) Ramakrishna, N. V. S., et al. "Rapid, simple and highly sensitive LC‐ESI‐MS/MS method for the quantification of tamsulosin in human plasma." Biomedical Chromatography 19.10 (2005): 709-719.

2) Chen, Lingdi, and Yu Zhang. "Determination of Mirabegron in rat plasma by UPLC–MS/MS after oral and intravenous administration." Revista da Associação Médica Brasileira 65 (2019): 141-148.